# Observational Study on the Prevalence of Urinary Incontinence in Female Athletes

**DOI:** 10.3390/ijerph18115591

**Published:** 2021-05-24

**Authors:** Jorge Velázquez-Saornil, Encarnación Méndez-Sánchez, Sonia Gómez-Sánchez, Zacarías Sánchez-Milá, Ester Cortés-Llorente, Ana Martín-Jiménez, Elena Sánchez-Jiménez, Angélica Campón-Chekroun

**Affiliations:** 1Department of Physiotherapy, Faculty of Health Sciences, Universidad Católica de Ávila, 05005 Ávila, Spain; encarnacion.mendez@ucavila.es (E.M.-S.); sonia.gomez@ucavila.es (S.G.-S.); zacarias.sanchez@ucavila.es (Z.S.-M.); ana.martin@ucavila.es (A.M.-J.); elena.sanchez@ucavila.es (E.S.-J.); angelica.campon@ucavila.es (A.C.-C.); 2FisioSalud Ávila, 05002 Ávila, Spain; estercll301296@hotmail.com

**Keywords:** urinary incontinence, women’s health, sports, incontinence, urinary stress

## Abstract

Purpose: To study the prevalence of UI in female athletes, the category of sports with the highest number of cases, the most incident risk factors and the bio-psycho-social consequences. Methods: Preparation of a survey, based on two validated questionnaires answered by 63 participants, to carry out an analytical, transversal and observational study. All participants were European, adult, female athletes (mean age 30.78 years, standard deviation 12.16 years). Results: UI has a high prevalence (44.4%) in female athletes (compared to 10% in non-athletes), being more frequent in those who practice long-distance running. As age and years of sport practice increase, the incidence of this pathology increases. Absorbent pads are used by more than half of women with incontinence, while the rest wet their underwear. Menopause, childbirth and surgery in the region are risk factors for UI, while the presence of urinary tract infections or candidiasis is not. The results state that urine loss does not cause anxiety or depression, but it does affect sporting life. Conclusions: The prevalence of UI in this study is very high and more common in female athletes and the incidence increases with age and other risk factors. The salient consequence of this study is that urine loss affects their sporting environment, but does not induce depression or anxiety.

## 1. Introduction

UI is one of the most common genitourinary problems found in women, along with pelvic organ prolapses (POPs), fecal incontinence and sexual dysfunction [1]. 

According to the International Continence Society (SCI), UI is defined as “the involuntary loss of urine, which can be observed and examined” [2]. This experience is a social taboo, understood as an embarrassing situation with negative impacts on social, working and emotional life [3]. Women with UI generally use containment measures, with absorption pads predominating [4].

The pelvic floor is a structure of muscles and connective tissue that provides support and suspension structures for the pelvic and abdominal organs. Its main component is the levator ani muscle, a muscle that covers most of the pelvis [1]. The pelvic organs can be divided into three compartments: anterior (bladder and urethra), middle (uterus and vagina, prostate and seminal vesicles) and posterior (rectum, anal canal and sphincteric apparatus) [2]. These structures have a close relationship with the pelvic floor musculature, which is involved in the functions of each of these. They provide not only mechanical support but also participate in urinary and fecal continence [1,2,3].

In a study of female students of physical education, it was observed that six out of seven of them presented stress UI. Therefore, the prevalence in female athletes is very high and adapting measures for the detection or prevention of UI in physical education and sport and knowing the mechanisms by which it is produced can help in the prevention and treatment of this pathology [5,6].

UI is not a life-threatening disease, but it does influence a person’s quality of life, diminishing it and causing a decline in mood along with anxiety and a worsening of social and sexual life [7,8,9]. 

### 1.1. Classification

#### 1.1.1. According to Sandvick’s Index, UI Can Be Classified, according to Its Severity, as

Mild.Moderate.Severe.Very serious [10,11,12].

#### 1.1.2. To Assess Which of the Three Is Incontinence, Two Questions Are Given to the Patient and according to the Answers, the Patient Is Classified as

-How often do you experience urine loss? Less than once a month.Once or more times a month.Once or more times a week.Every day and/or night.
-How much do you lose?
Drops (small amount).Small jet (moderate amount).A lot.

#### 1.1.3. After Answering, the Scores of the Results Are Multiplied and Classified as Follows

1–2: slight.3–6: moderate.8–9: severe.12: very serious.

#### 1.1.4. According to the SCI, UI Can Be Classified According to Symptomatic or Urodynamic Criteria, Which Are Explained Below

Stress urinary incontinence (SUI): the involuntary loss of urine associated with physical exertion that causes increased abdominal pressure, which overcomes urethral pressure to contain urine. The resistance mechanisms of the pelvic floor (musculature, ligaments, sphincters) are weak, so they do not contain urine [13,14]. These efforts can be coughing, sneezing, jumping and running.Emergency urinary incontinence (UTI): the involuntary loss of urine immediately preceded by an emergency situation, due to an involuntary contraction of the detrusor muscle of the bladder (hyperactivity of the detrusor) [13,15]. The person with this type of incontinence feels the urge to urinate suddenly and is not able to hold back the urine [13,16].Mixed urinary incontinence (MUI): the combination of the two previous types [13,15]. Urine loss is associated with both emergency situations and physical exertion.Continuous urinary incontinence (CUI): the loss of urine continuously and involuntarily. It may be due to various pathologies such as fistulas or intrinsic deficiencies of the urethra [13].Nocturnal enuresis: the involuntary loss of urine during the night, while sleeping.Unconscious urinary incontinence (IUI): the involuntary loss of urine caused unconsciously (without the desire to urinate) and without increasing abdominal pressure. It is very rare, occurs in the elderly, and occurs when the bladder is at its fullest [13,14,15,16].Of all the urinary incontinences known, the most frequent in sportswomen are SUI, MUI and UTI [14].

### 1.2. Epidemiology 

UI is two to four times more frequent in women than in men, so much so that its prevalence is between 20% and 40% and its annual incidence is 2% to 11% in countries such as Germany and the United Kingdom [17]. In Spain, the prevalence is around 23% (similar to Norway with 25%), while in other countries such as France, Germany and the United Kingdom, the value is 35%.

Of all the types of UI that appear in athletes, the most common is SUI, followed by MUI, while the least prevalent is UTI [6,18]. 

In the study by Alves et al. [7], UI was assessed among women aged 18–40, nulliparous women and sportswomen. The result was that 22.9% of them had UTIs and of this percentage, 60.7% were SUI and 25% UTI, followed by MUI at 14.3%. 

Generally, SUI is associated with younger and pre-menopausal women while, as age increases (and with menopause), the prevalence of UIs increases in parallel, especially MUI and UTI [3,10]. It is so high in geriatric ages that it is one of the most predominant syndromes. With all these data, it is surprising how few patients ask for a consultation for this reason [19]. 

Nygaard’s study [20] observed the percentages of UI as age increases with the following results: 7% in women between 20 and 39 years of age, 17% in the range of 40 to 59 years of age, 23% between 60 and 79 years of age and 32% in women over 80 years of age.

Despite the high prevalence of UI, very few people go to a specialist to address the problem, often because of embarrassment and taboo of the subject matter exposed. In sportswomen, this pathology is poorly reported and recognized, and undertreated [21]. 

In a study carried out with female athletes, it was observed that nearly 30% of them had urine loss, at least during some type of exercise. In addition, the prevalence is higher in athletes than in those who do not practice sports and especially in those who practice impact sports, especially those involving jumping [22].

The main objective of this study is to assess the prevalence of UI in federated female athletes (federative sports license). In addition to observing urine leakage in the study subjects according to the sport discipline of athletics and age, the methods of containment, the characteristics of leakage and the risk factors that may lead to UI were analyzed. The biopsychosocial component of UI in this group of athletes was also analyzed.

## 2. Materials and Methods

The study was conducted according to the guidelines of the Declaration of Helsinki and approved by the Ethics Committee of Hospital Nuestra Señora de Sonsoles (Ávila, 13 of February of 2020) and has the clinical register trial registration number (TRN) NCT0435264715-april-2020. All participants confirmed informed consent for the study.

### 2.1. Study Design

This was an analytical, cross-sectional, observational study. A cross-sectional study is defined as a type of observational research that analyzes variable data collected over a period of time for a sample population. This study was carried out through surveys disseminated through various media, mostly online and in person. The surveys were created using the Google Forms program. A total of sixty-three people answered the surveys. 

The survey contained twenty-six questions on basic data, risk factors, types and amount of UI and emotional impact of the pathology.

#### 2.1.1. Inclusion Criteria 

All the people who responded to the survey met the following requirements: Female SexOver 18 years of age and federated in athletics.Belonging to Castilla y León (Spanish autonomous community whose territory is located in the northern part of the plateau of the Iberian Peninsula and corresponds mostly to the Spanish part of the Duero river basin. It is made up of nine provinces: Ávila, Burgos, León, Palencia, Salamanca, Segovia, Soria, Valladolid and Zamora. It is the largest autonomous community in Spain, with a surface area of 94 226 km^2^, and the sixth most populated, with 2,409,166 inhabitants.

#### 2.1.2. Exclusion Criteria

Men.Those women who are no longer federated or who are under 18.

### 2.2. Preparation of the Survey 

The survey “UI in female federated athletes” was the means of collecting data used to do this study. This survey is based on the “King’s Health Questionnaire” in its Spanish version validated by Xavier Badía et al. [23] and on the “International Consultation on Incontinence Questionnaire-Urinary Short Form (ICQ-UI SF)” in its Spanish version validated by Espuña Pons, Rebollo Álvarez and Puig Clota [24]. The rest of the survey was prepared by the authors. The aim of the survey was to study the presence of UI in female federated athletics, its risk factors and consequences. 

The survey is considered qualitative and quantitative with questions based on reliable and valid questionnaires. The survey consists of 17 questions and can be divided into three parts:

The informative part, in which we can group the first 6 questions, besides 11, 12 and 13. These are questions of our own and 16 questions that aim to find the factors of risk and which their exact relation with athletics.

The part that studies UI, in which questions 7 to 10 are found, which is based on the “King’s Health Questionnaire”. These questions assess the type and degree of UI. 

The part that assesses the bio-psycho-social environment, questions 14 to 17, studies the daily social and psychological impact of UI in athletes. It also aims to analyze the quality of life of each of the respondents and is based on the questionnaire “Incontinence Questionnaire-Urinary Short Form (ICQ-UI SF). The aim was to ensure that the answers were accurate and provide as much information as possible. The survey is shown in Appendix A, together with the validity and reliability of the questionnaires on which the survey questions were based. 

The study subjects were women who are federated in athletics in all the provinces of Castilla y León, in total, 525 women. It was not necessary to exclude any responses, since without fulfilling the requirements the survey could not be answered. 

The IBM SPSS Statistics 27 software was used for the statistical analysis of the data obtained. The descriptive parameters were calculated for all the quantitative variables: mean and standard deviation. The qualitative variables are expressed by means of relative percentage frequencies. The comparison between variables was made by means of Pearson’s chi-square test (χ2) for categorical variables, and by means of a Student’s *t*-test for quantitative variables. This comparison was calculated in the 95% confidence interval, so the results were considered significant if the *p*-value was less than 0.05 (*p* < 0.05). 

## 3. Results

### 3.1. Results for Each Variable 

The average age of the women who answered the survey is 30.78 (with a standard deviation of 12.16), with the youngest being 18 (a requirement of the study) and the oldest 61. Finally, the average number of years practicing athletics is 15.35 (8.52), with the minimum value being 3 years and the maximum 39 years.

#### 3.1.1. Sex and Location

All respondents were women and belonged to the Castilla y León Athletics Federation, as it was a criterion for inclusion and a requirement for answering the survey. 

The responses of the sport discipline have been grouped into five categories: (ordered from most common to least common practice according to the responses).

Long-distance running. This category is made up of the 5000 and 10,000 m, the half marathon and the marathon; 27% of the women surveyed gave this answer. Long-distance races are considered to be those over 3000 m in distance and with endurance tests.Speed events. This includes responses from hurdles and sprint events of 100, 200 and 400 m, which constitute 23.8% of the responses.Middle distance races. With 19% of the total, this group includes 800, 1500 and 3000 m. This category includes all events with a distance range between 800 and 3000 m. These competitions combine speed and endurance.Jumping events. Pole vault, high jump and long jump are the answers that make up this group. Jumping events account for a total of 17.5% of the total.Throwing events. Include shot put, hammer throw, javelin and discus; 12.7% of the responses, the lowest percentage of all the events.

In this study, 70.3% of the athletes trained more than eight hours per week and 29.7% trained between three and five hours per week.

#### 3.1.2. UI

UI is quite common in female athletes (prevalence: 28 women versus 35). In the sample of 63 women, 44.4% of them reported urine leakage, which a priori is understood to be a high percentage. This high percentage of incontinence is to be taken into account, given that the majority of the sample population is young (mean age 30.78 years).

In terms of the frequency of leakage, none of the women reported leakage continuously during the day. The most unusual frequency of leakage was less than twice a week and between five and two times a week, with 53.3% and 35.5%, respectively. Taking these values into account, overall, in this sample, UI is classified as mild as leakage is infrequent.

Thirty-five women (55%) responded that they do not leak urine. Of the remaining 28 women who do leak, 37.3% described their leakage as minimal, “just droplets”. Only 6.3% referred to the amount of their leakage as moderate, while no women indicated the option of ‘‘heavy leakage’’ in the survey. These results describe the incontinence of these athletes as light, as the vast majority of them leak very little.

Of the 28 women with UI, 25% reported coughing and sneezing and 64.3% reported coughing and sneezing during physical exercise. If we add these two together, we get a total of 89.3%, which means that it can be assumed that the vast majority suffer from SUI. Few reported incontinence at the end of urination (3.6%) and none ticked the option “at night, during sleep” or “before going to the toilet”. There is a percentage (7.2%) who do not relate their incontinence to any occasion (or at least of the options shown).

Women claiming to have urinary incontinence account for 44.4%, but not all of them use means of containment. In addition, 17.4% report not using means of containment, but they do wet their underwear; the remaining 27% use panty liners either daily (12.7%) or occasionally (14.3%). Therefore, the use of means of containment is frequent among sportswomen, as more than half of the incontinent women use them, also taking into account that this is a young population sample.

#### 3.1.3. UI and Bio-Psycho-Social Component

-The women surveyed show how UI affects their daily lives. A total of 79.4% of the women claim to have no social or sexual problems, while the remaining 20.6% claim to have problems in the area of sport.-Anxiety, stress, depression. This psychological discomfort may be caused by the UI, as it is true that sometimes it can cause bad odor or stains on clothes, which cause concern among these women. In this study, very few women were affected by this, only 7.9%, which is equivalent to five women.

Table 1 shows the relationship of urinary incontinence with different variables, to see if these are statistically significant. On the one hand, the number of women with UI, “yes”, is shown with the exact number (“N”) and percentage (“%”) related to the variable to be crossed. On the other hand, “no” refers to the total number of women who do not experience UI, similar to the previous variable in terms of the number and percentage that appears in each case. The number of women who participated in this survey (63), incontinent or not (“total”), is also shown. A total of 28 women (44.4%) reported UI compared to 35 women who reported not suffering from UI (55.6%).

In terms of sporting discipline, there is no significant relationship with UI, but it is observed that the greatest number of athletes with UI practice long-distance running, represented by 32.1% of all women experiencing losses, followed by middle-distance running with 25% of cases. In contrast, jumping events account for the lowest percentage of incontinent women, with 10.7%, after sprinting (17.9%) and throwing events (14.3%). Leakage caused by exercise or physical exertion accounts for 64.3% and that caused by coughing or sneezing for 25%. A total of 58.6% of women use a napkin and 39.3% wet their underwear.

It is interesting to study when UI occurs in order to be able to narrow down the type of incontinence that exists. UI caused by exercise or physical activity stands out from the rest, with 64.3% (18 women). This is followed by UI caused by coughing or sneezing (25%). If we add these two together, the combined percentage would be 89.3% for SUI, which is the vast majority. There are no cases of nocturnal leakage, neither continuous nor before reaching the toilet, but there is one case of UI after having finished urinating. Some 7.2% of women do not relate their incontinence to any of these occasions.

Risk factors such as menopause, natural childbirth, surgery or associated disorders are described above. The first three have a significant relationship with urinary incontinence, as the *p*-value is less than 0.05. In the case of associated disorders, no significant relationship with urinary incontinence has been described, as the *p*-value is greater than this number (0.929).

As the population of this sample is young, it is to be expected that most of them have not entered the menopausal period (88.9%). Of the female athletes with UI, 25% are menopausal, which is related to the number of mature women in this sample.

Regarding childbearing, 23.8% of the women in this study had given birth at some point in their lives and 42.9% of them reported urine leakage, compared to 8.6% of women continents who had also given birth. A significant relationship was found between UI and childbirth.

A total of 77.8% of the sportswomen surveyed had not undergone any pelvic and/or abdominal surgery in their lifetime, but within the 22.2% of those who had entered the operating theatre, 35.7% are incontinent. Given these results, the *p*-value informs us of the significant relationship between these two variables.

Table 2 shows no significant relationship between suffering with associated disorders and UI. Most of the incontinent women in this sample did not describe any pathology associated with UI (89.3%).

Table 3 relates some of the possible consequences of UI.

Firstly, the aim was to study what type of effects this pathology may have on the daily lives of the sportswomen and it was found that 46.4% were affected in their sporting environment compared to 53.6% (also incontinent) who did not report any type of effect. Therefore, the only effect they described was in sporting, and they did not present problems in their social or sexual life, nor in their domestic environment. Therefore, a significant relationship was found between women who have urinary incontinence and those who are affected in the field of sports (*p*-value of 0.000).

When assessing the relationship between urine loss and suffering from anxiety and depression, no significant relationship was found, as the *p*-value was greater than 0.05 (0.096). The percentage of incontinent women with these consequences was very low, only 14.3% compared to 85.7% of incontinent women without these disorders.

## 4. Discussion

The results obtained by this study lead to the acceptance of the projected hypotheses, due to the fact that numerous cases of UI have been found in athletes, with a prevalence of 44.4%. It can be confirmed that there are different risk factors related to loss of urine and that the type of UI that defines female athletes is SUI. UI is more prevalent in sportswomen than in non-sportswomen [6,25,26].

Urine losses in athletes are related to the repetitive impact on the floor, which in the long term weakens the pelvic diaphragm in female and causes SUI [12]. UI is more prevalent in sportswomen than in non-sporting women [6,14,15].

Not all sports are impact sports, and of those that are, not all to the same degree of intensity, and those that are higher impact sports are more likely to cause UI [6]. In athletics, there are different disciplines, all of which are impact sports to varying degrees. Jumping events have been found to be the least prevalent in terms of UI, which was expected to be one of the most prevalent, because jumping is a major factor in weakening the pelvic floor [21]. Long-distance running, on the other hand, is more likely to be prevalent because of the long duration of impact. 

In this sample, most of the women were young, which led to the small number of cases of menopause, childbirth and surgery. However, all three have been found to be risk factors, as almost all women who have undergone all three have UI [25,26,27]. These factors may explain UI, as they cause alterations in the pelvic floor and eventually weaken it [27]. For example, the menopause reduces the secretion of estrogens [28], which are responsible for forming the collagen in the ligaments (which support the pelvic floor). The menopausal factor can be considered indirect but, on the other hand, childbirth and surgery can directly provoke alterations in the abdomino-pelvic area (producing soft tissue deficit and weakness) [28].

On the other hand, the literature states that the presence of urogenital diseases helped the development of UI [29], but this was not the case in this study because few women reported these associated pathologies. Furthermore, in a recent investigation by Pero et al. [30], in which 12 male athletes were studied, it is clear from the results obtained in this intervention that the prevention of new infections together with timely targeted antibiotic treatment by sports physicians for existing infections reduces the continuation of infections that can be aggravated by intense physical exercise, which is why it should be related to our study, since only 11% of the women reported an illness associated with UI. Therefore, urine leakage has not only a physical significance, but also a very powerful psychological one and even the risk of associated infections [29,30].

Generally, incontinent women report social, sexual and sports problems [19,22] but, in this case, the athletes only pointed out the effects on sports, with a significant relationship. This is probably due to the fact that the most important thing for them is their sporting performance.

Anxiety and depression are very common consequences for incontinent women [21], due to the embarrassment of being incontinent because of leaks or the use of pads [14]. In this case, a significant relationship was not found, which may be due to the fact that the participants were athletes and the sport itself creates a sense of well-being [2] and, in addition, female athletes may already have known the strong relationship between playing sports and suffering from UI.

It is a fact that age is a very powerful factor in UI [14,17,27,28,29], due to all the physical changes that occur in the body, in addition to all the stages that come with age. Age is therefore directly related to UI, but the low mean age in this study and the number of cases of UI are noteworthy.

Sporting activities gradually weaken the pelvic floor over time, due to the impact produced [6,14,19]. Therefore, the more years of athletic training, the greater the rate of incontinence.

### Limitations and Strengths

The main limitation of this study is the small sample size. A larger number of participants would have been preferable, but the current timing of the pandemic limited the study. Another limitation is that the survey was exclusive to federated athletes, so other non-federated female athletes could not respond to the survey. In addition, 100% of the respondents were female, but this cannot be verified due to the fact that the survey was completed online, due to the COVID-19 pandemic situation, although they were required to provide their federation license number by which the identity of the subject who completed the survey was verified. In addition, several variables have been studied in detail, which is considered a strength of the study.

## 5. Conclusions

The prevalence of UI in the federated athletes in this study is high, in 44.4% of cases, and the incidence increases with age and other risk factors such as surgery, childbirth or menopause.

It has been found that the sporting discipline that most damages the pelvic floor musculature is long-distance running and the least damaging is jumping. UI in sportswomen is mild and SUI is the most common in this study (89.3%), in addition to exercise UI (64,3%) and UI due to coughing or sneezing (25%), but UI to more severe degrees is low in this study. In addition, more than half of the women with incontinence use pads on a daily or occasional basis. The most striking implication of this study is that urine leakage affects their sporting performance, but does not induce depression or anxiety. 

Further studies are needed to demonstrate the prevalence of UI in female athletes, so that prevention and intervention programs can be implemented for these athletes to reduce the prevalence of UI.

## Figures and Tables

**Table 1 ijerph-18-05591-t001:** Relationship of UI.

	Yes N Total = 28 *N (%)*	No N Total = 35 *N (%)*	Total N Total = 63 *N (%)*	*p*
**Athletic discipline**				0.46
Speed tests	5 (17.9)	10 (28.6)	15 (23.8)	
Throwing tests	4 (14.3)	4 (11.4)	8 (12.7)	
Jump tests	3 (10.7)	8 (22.9)	11 (17.5)	
Long-distance running	9 (32.1)	8 (22.9)	17 (27)	
Middle-distance running	7 (25)	5 (14.3)	12 (19)	
**Chance losses**				0.00
Never	0 (0)	35 (100)	35 (56.5)	
Cough/Sneeze	7 (25)	0 (0)	7 (11.1)	
Physical efforts	18 (64.3)	0 (0)	18 (28.6)	
Before going to the bathroom	0 (0)	0 (0)	0 (0)	
Finish urinating	1 (3.6)	0 (0)	1 (1.6)	
Continuously	0 (0)	0 (0)	0 (0)	
At night	0 (0)	0 (0)	0 (0)	
No obvious reason	2 (7.2)	0 (0)	2 (3.2)	
**Loss frequency**				0.00
Continuously	0 (0)	0 (0)	0 (0)	
Several times/day	1 (3.6)	0 (0)	1 (1.6)	
One time/day	1 (3.6)	0 (0)	1 (1.6)	
More tan five times/week	2 (7.1)	0 (0)	2 (3.2)	
Two to five times/week	10 (35.5)	0 (0)	10 (15.9)	
Less than twice/week	15 (53.3)	0 (0)	15 (23.8)	
No losses	(0)	35	35 (52.4)	
		(100)		
**Means of containment**				0.00
Daily	8 (27.1)	0 (0)	8 (12.7)	
Occasionally	9 (31.5)	0 (0)	9 (14.3)	
No, but wet underwear	11 (39.3)	0 (0)	11 (17.5)	
No losses	0 (0)	35(100)	35 (55.6)	
**Total**	28 (44.4)	35	63 (100)	
		(55.6)		

**Table 2 ijerph-18-05591-t002:** UI and risk factors.

	Yes *N (%)*	No *N (%)*	Total *N (%)*	*p*
**Menopause**				0.002
Yes	7 (25)	0 (0)	7 (11.1)	
No	21 (75)	35 (100)	56 (88.9)	
**Births**				0.001
Yes	12 (42.9)	3 (8.6)	15 (23.8)	
No	16 (57.1)	32 (91.4)	48 (76.2)	
**Surgeries**				0.021
Yes	10 (35.7)	4 (11.4)	14 (22.2)	
No	18 (64.3)	31 (88.6)	49 (77.8)	
**Associated disorders**				0.929
Yes	3 (10.7)	4 (11.4)	7 (11.1)	
No	25 (89.3)	31 (88.6)	56 (88.9)	
**N Total**	28 (44.4)	35 (55.6)	63(100)	

**Table 3 ijerph-18-05591-t003:** UI and consequences.

	Yes*N (%)*	No*N (%)*	Total*N (%)*	*p*
**Affect on daily life**				0.000
Yes, sports field	13 (46.4)	0 (0)	13 (20.6)	
Yes, social life	0 (0)	0 (0)	0 (0)
Yes, domestic	0 (0)	0 (0)	0 (0)
Yes, sex field	0 (0)	0 (0)	0 (0)
No	15 (53.6)	35 (100%)	50 (79.4)
**Anxiety, depression**				0.095
Yes	4 (14.3)	1 (2.9)	5 (7.9)	
No	24 (85.7)	34 (97.1)	58 (92.1)
**N total**	28 (44.4)	35 (55.6)	63(100)

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
