# Peer review of "Observational Study on the Prevalence of Urinary Incontinence in Female Athletes"

_ijerph, 2021, doi:10.3390/ijerph18115591_

Round 1

Reviewer 1 Report

I can read the review versión and the manuscript is ok.

Author Response

Thank you very much for your comments, it encourages us to continue our work.

Best regards.

Reviewer 2 Report

Dear authors. The manuscript has improved. Ready to publish. Congratulations. Greetings

Author Response

(The authors gave the same response as above.)

Reviewer 3 Report

There is a major flaw in the research design. In this study, it is needed to add control group because the great risk factor of UI is age.

Overall, it needs to be modified to fit the format of the article.(especially, Line 50-93).

Author Response

After the last review of the paper by the three reviewers, we have received very good reviews from two reviewers, but there is one allegation from you that we do not agree with. He comments: "There is a major flaw in the research design. In this study, it is necessary to add a control group because the major risk factor for UTI is age". From our point of view, this is a descriptive study in which we limit ourselves to describing the population with pathology and with specific characteristics, without the need to compare with a healthy group or control group, which is a great contribution for a future study. However, for the present study we understand that we have described the epidemiology of incontinence with incidence and prevalence in different countries and we are only looking at the prevalence of incontinence in a certain group. Observational studies are those in which the researchers act as observers, without any influence on the exposure factors, our research team has asked questions through a questionnaire that has been answered by a specific group of people and we have limited ourselves to study these results, without any intervention. We understand that applying a control group to our study would change the study design significantly, and we welcome your input to complement our research for a future manuscript, but we understand that this observational study has the correct design. Please understand our point of view, otherwise the study would change drastically. In addition, we have modified the sections and subsections of the article, especially on pages 50-97, to comply with the journal's guidelines as stated in the formatting template.
 Thank you very much for your contributions.

This manuscript is a resubmission of an earlier submission. The following is a list of the peer review reports and author responses from that submission.

Round 1

Reviewer 1 Report

Line 111. The aim of the study is very general, should be better define.

Line 130. "Over 18 years of ageFederated in athletics". Review grammar. Define Athletics ( all the disciplines was incluyed?).

Line 140. Reference 24 must be modificated in the text.

lines 154 to 164 would be more correct if it´s into "Material and Methods" since it is part of the development of the study and is not a result variable.

A brief clarification  of the statistical analysis is advised in the material section

Author Response

Line 111. The objective of the study is very general, it should be better defined. 
The objective has been modified and focused on more specific aspects of this study.

Line 130. "Over 18 years oldFederate in athletics". Review grammar. Define Athletics (were all disciplines included?).
Corrected sentence, thank you. All disciplines have been included and are now detailed in the text.

Line 140. Reference 24 should be amended in the text. 
This reference has been amended, thank you.

Lines 154 to 164 would be more correct if it were "Material and Methods" as it is part of the development of the study and is not an outcome variable. 
A brief clarification of the statistical analysis in the material section is advised. 
Your comment has been taken into account, thank you. In addition, the statistical analysis has been expanded below the tables.

Reviewer 2 Report

Thank you for asking me to review the manuscript entitled "Observational study on the prevalence of urinary incontinence 2 in federated athletes.". This paper properly discuss the relevant literature to the topic, and the methodological approach sounds correct for the design of the study.

I would like to suggest to:

The title should add "famale".

I would like to suggest to first define in the introduction the meaning of "UI" (although already defined in the abstract, that should also be done in the main text).

Line 68. The introduction should add a paragraph about pelvic floor justifying this objective.

Line 77. Explain the difference between urinary incontinence (UI) and continuous urinary incontinence (UI).

Line 118. Materials and Methods. Study design.

It is necessary to add in this section the fulfillment of the Helsinki declaration, the ethical treatment and the number of the local ethics committee.

Line 131. Why it is considered an inclusion criterion belonging to Castilla y León (community of Spain)? Is a important region?

Line 343: Error in reference 24

Author Response

The title should add "famale".

The word female has been added, thank you.

I would like to suggest to first define in the introduction the meaning of "UI" (although already defined in the abstract, that should also be done in the main text).

UI has been defined according to the International Continence Society (ICS), line 28.

Line 68. The introduction should add a paragraph about pelvic floor justifying this objective.

A specific paragraph on the pelvic floor has been added (line 34-42), good input, thank you.

Line 77. Explain the difference between urinary incontinence (UI) and continuous urinary incontinence (UI).

The differences between the different types of incontinence have been explained, lines 75-96.

Line 118. Materials and Methods. Study design. It is necessary to add in this section the fulfillment of the Helsinki declaration, the ethical treatment and the number of the local ethics committee.

A specific section on ethical treatment has been added in this section, as requested.

Line 131. Why it is considered an inclusion criterion belonging to Castilla y León (community of Spain)? Is a important region?

Castilla y León is a Spanish autonomous community whose territory is located in the northern part of the plateau of the Iberian Peninsula and corresponds mostly to the Spanish part of the Duero river basin. It is made up of nine provinces: Ávila, Burgos, León, Palencia, Salamanca, Segovia, Soria, Valladolid and Zamora. It is the largest autonomous community in Spain, with a surface area of 94 226 km², and the sixth most populated, with 2 409 166 inhabitants. Reviewer 2 has commented on whether it was an important region and that is why it is clarified in this comment, but we understand that it is not necessary to introduce it in the text, as the geographical term is not important in this speciality.

Line 343: Error in reference 24.

Corrected this reference, thanks for the note.

Reviewer 3 Report

Thank you for the opportunity to review this paper. I hope my comments are helpful.

General queries:

What is a federated athlete? please define.

Abstract: you state the study is transversal. Is this 'transverse'? this is the same as cross-sectional. There needs to be a few lines on the significance of the study in the abstract. Your abstract findings need to include a comparison with the general population continence rates for women.

Introduction:

Reword 'women who attend courses' sentence- I am unsure what you are referring to here. What type of courses? your discussion that incontinence may be higher in athletes than non athletes needs to be drawn more strongly from reference 6 and this needs to be a major part of the significance of your study.

You don't need to include so much description o the types of urinary incontinence. Miss out the types not covered in your own study - such as nocturnal enuresis.

Epidemiology- context needed- which countries is UI 2 to 4 times more common in women? globally? [ref 17]. The relevance of much of the epidemiology discussion is unclear- such as why women do not always seek help for incontinence , and the discussion on older women.

Your objectives do not fully reflect those in the abstract.

Materials and methods

I am unsure how your study can be cross sectional and retrospective- please explain. survey- explain please how the authors prepared the rest of the survey? which evidence was used to inform it? how?

What do you mean by the beneficiaries of the study? how do these women benefit?

What is meant by qualitative variables? were these open ended questions? were they thematically analysed? did you use content analysis? - needs explaining.

The athletic disciplines of the participants may need re explaining. I don't know what launch tests means- for example.

UI results: what do you mean by the average is 30.78 years

Overall the findings section needs to be organised and explained much more clearly. You seem to draw conclusions not supported by the evidence- for example, your findings on anxiety stress and depression. Please refer to tables when discussing findings.

there is no limitations section or ethical issues discussion in your paper.

Table 2. is this meant to say menopause (minuspause).

Some of your findings- such as the relationship between childbirth a, age and incontinence are well known. Focus on what new findings you have.

Discussion

Some of this section draws unsafe conclusions. The discussion must consider a range of reasons behind the findings based on past research. stress what is different in your findings.

Conclusions- you cannot really state that UI is high in athletes unless you compare to the general population.'You include new findings in your conclusion.  I am unsure how you decide years of training influence UI prevalence but hours of weekly training do not.

You need stronger recommendations in your conclusion. You also need to summarise your main finding- was the research question answered?

Author Response

What is a federated athlete? please define. 

A federated athlete is an athlete with a federative sports licence and therefore can compete in different competitions at national and international level. 

Abstract: you state the study is transversal. Is this 'transverse'? this is the same as cross-sectional. There needs to be a few lines on the significance of the study in the abstract. Your abstract findings need to include a comparison with the general population continence rates for women.

Your comment has been taken into account and a comparative incidence of incontinence in the general population of women has been added. We consider this to be a cross-sectional study as these are defined as a type of observational research that analyses data on variables collected over a period of time on a pre-defined sample population or subset. This type of study is also known as a cross-sectional study, cross-sectional study and prevalence study. The data collected in a cross-sectional study comes from people who are similar.

Introduction:

Reword 'women who attend courses' sentence- I am unsure what you are referring to here. What type of courses? your discussion that incontinence may be higher in athletes than non athletes needs to be drawn more strongly from reference 6 and this needs to be a major part of the significance of your study.

Your comment has been taken into account. This sentence in line 32 has been corrected. In addition, your request to address incontinence in athletes versus non-athletes has been addressed and citation 6 has been referenced as you requested.

You don't need to include so much description o the types of urinary incontinence. Miss out the types not covered in your own study - such as nocturnal enuresis.

We think it is interesting to introduce this classification of the different types of incontinence and have been praised by some editors.

Epidemiology- context needed- which countries is UI 2 to 4 times more common in women? globally? [ref 17]. The relevance of much of the epidemiology discussion is unclear- such as why women do not always seek help for incontinence , and the discussion on older women. Your objectives do not fully reflect those in the abstract.

 Countries with the highest incidence of incontinence, line 98, have been included. Your comment has been taken into account and reworded, lines 98-115.

Materials and methods

I am unsure how your study can be cross sectional and retrospective- please explain. survey- explain please how the authors prepared the rest of the survey? which evidence was used to inform it? how?

The term retrospective has been removed in case there was controversy. The manuscript makes it clear how the survey was prepared and how it was answered. Lines 153-171.

What do you mean by the beneficiaries of the study? how do these women benefit? This term has been removed and replaced with a more precise term, line 172.

What is meant by qualitative variables? were these open ended questions? were they thematically analysed? did you use content analysis? - needs explaining.

A qualitative variable is a type of statistical variable that describes the qualities, circumstances or characteristics of an object or person, without making use of numbers. In this way, qualitative variables make it possible to express a non-numerical characteristic, attribute, quality or category. They are closed questions and can be checked in the attached supplementary material, although not publishable, they have been added so that they can be evaluated. In addition, the answers and their content were analysed. A document is attached to this review so that it can be evaluated, please see the attachment.

The athletic disciplines of the participants may need re explaining. I don't know what launch tests means- for example.

The different sport disciplines have been explained to make it more concise, lines 191-203, thank you for your comment.  The throwing events are: shot put, hammer, javelin and discus. 

UI results: what do you mean by the average is 30.78 years

This is the mean age of the women who participated in the study. Line 184

Overall the findings section needs to be organised and explained much more clearly. You seem to draw conclusions not supported by the evidence- for example, your findings on anxiety stress and depression. Please refer to tables when discussing findings.

Your comment has been taken into account and the manuscript has been modified. Specifically, the information has been detailed below the tables. The results refer to the findings in the tables, thank you for your input.

there is no limitations section or ethical issues discussion in your paper.

A limitations and strengths section has been included in the discussion (line 329-337). In addition, an ethical issues section has been included in line 130-133 as suggested by another reviewer to be included in this section and, in addition, in line 374-377 as indicated by the editors.

Table 2. is this meant to say menopause (minuspause).

This term has been modified, line 248, thanks to the reviewers.

Some of your findings- such as the relationship between childbirth a, age and incontinence are well known. Focus on what new findings you have.

Your comment has been taken into account and these findings have been discussed and highlighted in the results section.

Discussion

Some of this section draws unsafe conclusions. The discussion must consider a range of reasons behind the findings based on past research. stress what is different in your findings.

Your comment has been taken into account and this section has been modified and this discussion has been justified with previous research and based on scientific evidence. In addition, several more references have been added to make the discussion more robust.

Conclusions- you cannot really state that UI is high in athletes unless you compare to the general population.'You include new findings in your conclusion.  I am unsure how you decide years of training influence UI prevalence but hours of weekly training do not.

The conclusion that years of training influences UI prevalence but hours of weekly training has not been removed so as not to confuse the reader and the conclusions have been modified based on the findings shown in the study. UI is high in our study sample, as can be seen in table 1.

You need stronger recommendations in your conclusion. You also need to summarise your main finding- was the research question answered?

Your comment has been taken into account and the conclusions have been modified in the manuscript. Thank you for your comments. We have taken all of them into account in order to present a quality study and we hope that this will be the case. 

Round 2

Reviewer 3 Report

The article remains very poorly written. There is no significance or recommendations. Unnecessary content is included. Statistical analysis is carried out on tiny samples and conclusions drawn from them that are not justified. Recommended revisions are not addressed.